# Efficient Dictionary Learning with Switch Sparse Autoencoders

**Anish Mudide** [*]
Massachusetts Institute of Technology

**Joshua Engels**
Massachusetts Institute of Technology

**Eric J. Michaud**
Massachusetts Institute of Technology

**Max Tegmark**
Massachusetts Institute of Technology

**Christian Schroeder de Witt**
University of Oxford

## Abstract

Sparse autoencoders (SAEs) are a recent technique for decomposing neural network activations into human-interpretable features. However, in order for SAEs to identify all features represented in frontier models, it will be necessary to scale them up to very high width, posing a computational challenge. In this work, we introduce **Switch Sparse Autoencoders**, a novel SAE architecture aimed at reducing the compute cost of training SAEs. Inspired by sparse mixture of experts models, Switch SAEs route activation vectors between smaller "expert" SAEs, enabling SAEs to efficiently scale to many more features. We present experiments comparing Switch SAEs with other SAE architectures, and find that Switch SAEs deliver a substantial Pareto improvement in the reconstruction vs. sparsity frontier for a given fixed training compute budget. We also study the geometry of features across experts, analyze features duplicated across experts, and verify that Switch SAE features are as interpretable as features found by other SAE architectures.

## 1 Introduction

Recently, large language models have achieved impressive performance on many tasks (Brown et al., 2020), but they remain largely inscrutable to humans. Mechanistic interpretability aims to open this metaphorical black box and rigorously explain model computations (Olah et al., 2020). Specifically, much work in mechanistic interpretability has focused on understanding *features*, the specific human-interpretable variables a model uses for computation (Olah et al., 2020; Park et al., 2023; Engels et al., 2024).

Early mechanistic attempts to understand features focused on neurons, but this work was stymied by the fact that neurons tend to be *polysemantic*: they are frequently activated by several completely different types of inputs, making them hard to interpret (Olah et al., 2020). One theory for why neurons are polysemantic is *superposition*, the idea that language models represent many more concepts than they have available dimensions (Elhage et al., 2022). To minimize interference, the superposition hypothesis posits that features are encoded as almost orthogonal directions in the model's hidden state space.

Bricken et al. (2023) and Cunningham et al. (2023) propose to disentangle these superimposed model representations into monosemantic features by training unsupervised sparse autoencoders (Lee et al., 2007; Le, 2013; Konda et al., 2014) on intermediate language model activations. The success of this technique has led to an explosion of recent work (Templeton et al., 2024; Gao et al., 2024) that has focused on scaling sparse autoencoders to frontier language models such as Claude 3 Sonnet (Anthropic, 2024a) and GPT-4 (Achiam et al., 2023). Despite scaling SAEs to 34 million features, Templeton et al. (2024) estimate that there likely remains orders of magnitude more features. Furthermore, Gao et al. (2024) train SAEs on a series of language models and find that larger

---

[*]Correspondence to amudide@mit.edu.

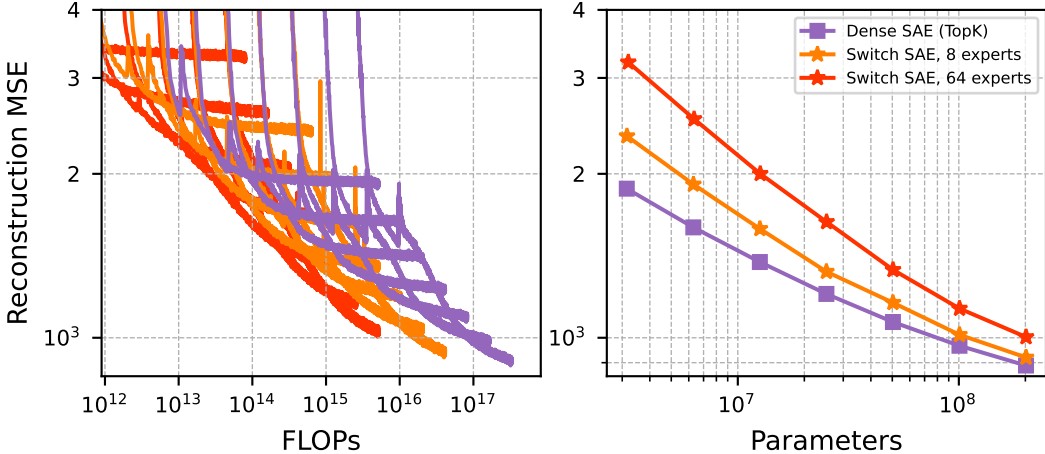

Figure 1: Scaling laws for Switch SAEs. We train dense TopK SAEs and Switch SAEs of varying size with fixed $k = 32$. **Left:** Switch SAEs achieve better reconstruction than dense SAEs at a fixed compute budget. **Right:** Switch SAEs require more features in total (and therefore more parameters) to achieve the same reconstruction as dense SAEs when trained to convergence, although this gap narrows for larger SAEs.

models require more features to achieve the same reconstruction error. As model sizes continue to grow, current training methodologies are set to quickly become computationally intractable.

Each training step of a sparse autoencoder generally consists of six major computations: the encoder forward pass, the encoder gradient, the decoder forward pass, the decoder gradient, the latent gradient and the pre-bias gradient. Gao et al. (2024) introduce kernels that leverage the sparsity of the TopK activation function to dramatically optimize all computations *except* the encoder forward pass, which is not sparse. After implementing these optimizations, Gao et al. (2024) find that the training time is bottlenecked by the dense encoder forward pass and the memory is bottlenecked by storing the latent pre-activations.

In this work, we introduce the Switch Sparse Autoencoder, which to our knowledge is the first work to solve these dual memory and FLOP bottlenecks. The Switch SAE combines the Switch layer (Fedus et al., 2022) with the TopK SAE (Makhzani & Frey, 2013; Gao et al., 2024). At a high level, the Switch SAE is composed of many small expert SAEs and a trainable routing network that determines which expert SAE will process a given input. We demonstrate that the Switch SAE delivers an improvement in training FLOPs vs. training loss over existing methods.

## 1.1 CONTRIBUTIONS

1. In Section 3, we describe the Switch Sparse Autoencoder architecture and compare it to existing SAE architectures. We also describe our training methodology, which balances reconstruction and expert utilization.

2. In Section 4.1, we describe scaling laws for reconstruction MSE with respect to FLOPs and parameters. We show that Switch SAEs achieve a lower MSE compared to TopK SAEs using the same amount of training compute.

3. In Section 4.2, we benchmark Switch SAEs against ReLU, Gated and TopK SAEs on the sparsity-reconstruction Pareto frontier.

4. In Section 4.3, we study feature duplication in Switch SAEs and visualize the global structure of Switch SAE features using t-SNE.

5. In Section 4.4, we demonstrate that Switch SAE features are as interpretable as TopK SAE features.

## 2 RELATED WORK

### 2.1 MIXTURE OF EXPERT MODELS

In a standard deep learning model, every parameter is used for every input. An alternative approach is conditional computation, where only a subset of the parameters are active depending on the input, allowing models with more parameters without the commensurate increase in computational cost. Jacobs et al. (1991) propose to train multiple networks, where each network is dedicated to a disjoint subset of all possible inputs. Since this seminal work on mixture-of-experts, significant follow-up work has been dedicated to exploring different architectures and configurations (Jordan & Jacobs, 1994; Tresp, 2000; Collobert et al., 2001; Rasmussen & Ghahramani, 2001; Aljundi et al., 2017). Shazeer et al. (2017) introduce the Sparsely-Gated Mixture-of-Experts (MoE) layer, the first general purpose conditional computation architecture component. A Mixture-of-Experts layer consists of (1) a set of expert networks and (2) a routing network that determines which experts should be active on a given input. Shazeer et al. (2017) use MoE to scale LSTMs to 137 billion parameters, surpassing the performance of previous dense models on language modeling and machine translation benchmarks. Building on this work, Fedus et al. (2022) introduce the Switch layer, a simplification to the MoE layer which routes to just a single expert and thereby decreases computational cost and increases training stability. Fedus et al. (2022) use Switch layers in place of MLP layers to scale transformers to over a trillion parameters. Recent state of the art language models have used MoE layers, including Mixtral 8x7B (Jiang et al., 2024) and Grok-1 (xAI, 2024). To the best of our knowledge, we are the first to apply conditional computation to training SAEs.

### 2.2 DEEP LEARNING TRAINING OPTIMIZATIONS

Our method fits into the wider literature focused on accelerating deep learning training. One type of training speedup uses hardware accelerators like GPUs (Raina et al., 2009) and TPUs (Jouppi et al., 2017) to optimize highly parallelizable dense matrix operations. Algorithmic improvements, on the other hand, consist of architectural or implementation tricks to speed up forward and backwards passes on fixed hardware. Techniques such as MoE layers (discussed above) and Slide (Chen et al., 2020) utilize sparsity to only evaluate a subset of the parameters for a given wide MLP layer. Other techniques such as Flash Attention (Dao et al., 2022) and Reformers (Kitaev et al., 2020) use hashing or structured matrices to reduce the time complexity of a transformer's attention mechanism. See Appendix A in Dao et al. (2022) for a comprehensive overview of the literature on algorithmic training optimizations. Our work differs from these papers in that we are concerned with not only whether the training optimization results in a speedup, but also whether SAE quality is preserved.

### 2.3 SPARSE AUTOENCODERS AND IMPROVEMENTS

Sparse dictionary learning is the task of decomposing input data into a sparse linear combination of basic elements called *atoms*, which together form a *dictionary* (Olshausen & Field, 1997; Elad, 2010). There exist a wide variety of techniques for performing dictionary learning, such as the method of optimal directions (Engan et al., 1999) and K-SVD (Aharon et al., 2006). However, such methods are not scalable to large language models and may not be faithful to the models internals. As such, recent work has focused on applying sparse autoencoders (Lee et al., 2007; Le, 2013; Konda et al., 2014), a simple approximation of sparse dictionary learning, to language models.

Since the initial works proposing SAEs to separate model representations (Cunningham et al., 2023; Bricken et al., 2023), there have been many proposed improvements to the SAE architecture. These have included alternative activation functions like ProLu (Taggart, 2024), TopK (Makhzani & Frey, 2013; Gao et al., 2024), and Batch-TopK (Bussmann et al., 2024), architectural changes to solve feature suppression caused by the L1 penalty (Rajamanoharan et al., 2024a;b), and changes to the optimization objective itself (Braun et al., 2024). There are many metrics along which these papers evaluate their improvements, but the most common metric is a Pareto frontier of SAE latent sparsity (measured as average L0), mean squared error, and feature interpretability; thus, these are the metrics we focus on in this paper. We also compare primarily against TopK SAEs (Gao et al., 2024) as our baseline, as recent work (Anthropic, 2024c) has shown that these achieve state of the art results on these metrics.

## 3 THE SWITCH SPARSE AUTOENCODER

### 3.1 BASELINE SPARSE AUTOENCODER

Sparse autoencoders are trained to reconstruct language model activations $\mathbf{x} \in \mathbb{R}^d$ by decomposing them into sparse linear combinations of $M \gg d$ unit-length feature vectors $\mathbf{f}_1, \mathbf{f}_2, \ldots, \mathbf{f}_M \in \mathbb{R}^d$. SAE architectures consist of an encoder $\mathbf{W}_{\text{enc}} \in \mathbb{R}^{M \times d}$, a decoder $\mathbf{W}_{\text{dec}} \in \mathbb{R}^{d \times M}$, bias terms (e.g., $\mathbf{b}_{\text{pre}} \in \mathbb{R}^d$) and an activation function. The TopK SAE (Gao et al., 2024) outputs a reconstruction $\hat{\mathbf{x}}$ of the input activation $\mathbf{x}$, given by

$$\mathbf{z} = \text{TopK}(\mathbf{W}_{\text{enc}}(\mathbf{x} - \mathbf{b}_{\text{pre}})) \tag{1}$$
$$\hat{\mathbf{x}} = \mathbf{W}_{\text{dec}}\mathbf{z} + \mathbf{b}_{\text{pre}} \tag{2}$$

The latent vector $\mathbf{z} \in \mathbb{R}^M$ represents how strongly each feature is activated. Since $\mathbf{z}$ is sparse, the decoder forward pass can be optimized by a suitable kernel. The loss is the reconstruction error $\mathcal{L} = \|\mathbf{x} - \hat{\mathbf{x}}\|_2^2$.

We additionally benchmark against the ReLU SAE (Anthropic, 2024b) and the Gated SAE (Rajamanoharan et al., 2024a). The ReLU SAE uses the ReLU activation function and applies an L1 penalty to the feature activations to encourage sparsity. The Gated SAE avoids activation shrinkage (Wright & Sharkey, 2024) by separately determining which features should be active and how strongly activated they should be.

### 3.2 SWITCH SPARSE AUTOENCODER ARCHITECTURE

The Switch SAE consists of $N$ expert SAEs $\{E_i\}_{i=1}^N$ as well as a routing network that determines which expert SAE should be used for a given input (Fig. 2). Each expert SAE $E_i$ resembles a TopK SAE with no bias term:

$$E_i(\mathbf{x}) = \mathbf{W}_i^{\text{dec}}\text{TopK}(\mathbf{W}_i^{\text{enc}}\mathbf{x}) \tag{3}$$

The router, defined by trainable parameters $\mathbf{W}_{\text{router}} \in \mathbb{R}^{N \times d}$ and $\mathbf{b}_{\text{router}} \in \mathbb{R}^d$, computes a probability distribution $\mathbf{p}(\mathbf{x}) \in \mathbb{R}^N$ over the $N$ experts given by $\mathbf{p}(\mathbf{x}) = \text{softmax}(\mathbf{W}_{\text{router}}(\mathbf{x} - \mathbf{b}_{\text{router}}))$. We route the input $\mathbf{x}$ to the most probable expert $i^*(\mathbf{x}) = \arg\max_i p_i(\mathbf{x})$. The output $\hat{\mathbf{x}}$ is given by,

$$\hat{\mathbf{x}} = p_{i^*(\mathbf{x})}E_{i^*(\mathbf{x})}(\mathbf{x} - \mathbf{b}_{\text{pre}}) + \mathbf{b}_{\text{pre}}. \tag{4}$$

The Switch SAE thus avoids the dense $\mathbf{W}_{\text{enc}}$ matrix multiplication by leveraging conditional computation. When comparing Switch SAEs to existing SAE architectures, we evaluate two cases: (1) FLOP-matched Switch SAEs, which perform the same number of FLOPs per activation but increase the number of features by a factor of $N$, and (2) width-matched Switch SAEs, which reduce the FLOPs per activation by a factor of $N$ while keeping the number of features constant.

### 3.3 SWITCH SPARSE AUTOENCODER TRAINING

We train the Switch SAE end-to-end. When computing $\hat{\mathbf{x}}$, we weight $E_{i^*(\mathbf{x})}(\mathbf{x} - \mathbf{b}_{\text{pre}})$ by $p_{i^*(\mathbf{x})}$ to make the router differentiable. We adopt many of the training strategies described in Templeton et al. (2024) and Gao et al. (2024) (see Appendix A.1 for details).[1]

The TopK SAE enforces sparsity via its activation function and thus directly optimizes the reconstruction MSE. Following Fedus et al. (2022), we train our Switch SAEs using a weighted combination of the reconstruction MSE and an auxiliary loss which encourages the router to send an equal number of activations to each expert to reduce overhead. For a batch $\mathcal{B}$ with $T$ activations, the auxiliary loss is given by $\mathcal{L}_{\text{aux}} = N \cdot \sum_{i=1}^N f_i \cdot P_i$. $f_i$ represents the proportion of activations that are routed to expert $i$, and $P_i$ represents the proportion of router probability that is assigned to expert $i$. Formally, $f_i = \frac{1}{T}\sum_{\mathbf{x} \in \mathcal{B}} \mathbb{1}\{i^*(\mathbf{x}) = i\}$ and $P_i = \frac{1}{T}\sum_{\mathbf{x} \in \mathcal{B}} p_i(\mathbf{x})$. The auxiliary loss is minimized when the batch of activations is routed uniformly across the $N$ experts. The reconstruction loss is defined to be $\mathcal{L}_{\text{recon}} = \frac{1}{T}\sum_{\mathbf{x} \in \mathcal{B}} \|\mathbf{x} - \hat{\mathbf{x}}\|_2^2$. Note that $\mathcal{L}_{\text{recon}} \propto d$. Let $\alpha$ represent a tunable load balancing hyperparameter that scales the auxilliary loss. The total loss $\mathcal{L}_{\text{total}}$ is given by $\mathcal{L}_{\text{total}} = \mathcal{L}_{\text{recon}} + \alpha \cdot d \cdot \mathcal{L}_{\text{aux}}$. We optimize $\mathcal{L}_{\text{total}}$ using Adam (Kingma, 2014). We find that results are not overly sensitive to the value of $\alpha$, but that $\alpha = 3$ is a reasonable default based on a hyperparameter sweep (see Appendix A.2 for details).

---

[1] Our code can be found at `https://github.com/amudide/switch_sae`

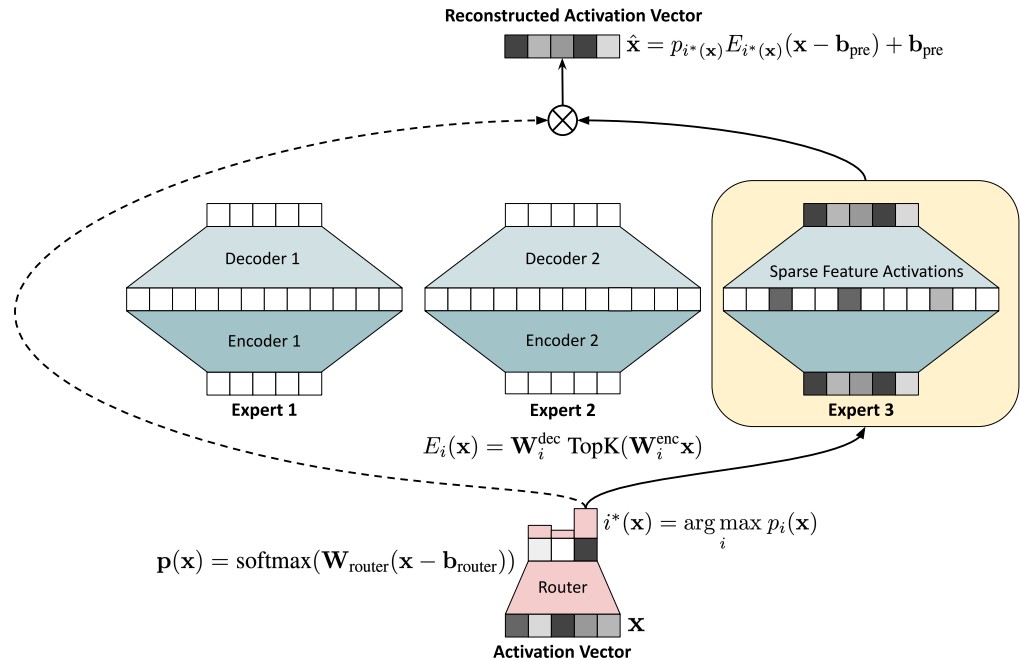

Figure 2: Switch Sparse Autoencoder Architecture. The router computes a probability distribution over the expert SAEs and routes the input activation vector to the expert with the highest probability. The figure depicts the architecture for $d = 5$, $N = 3$, $M = 12$.

# 4 RESULTS

We train sparse autoencoders on the residual stream activations of GPT-2 small, which have a dimension of 768 (Radford et al., 2019). Early layers of language models handle detokenization and feature engineering, while later layers refine representations for next-token prediction (Lad et al., 2024). In this work, we follow Gao et al. (2024) and train SAEs on activations from layer 8, which we expect to hold rich feature representations not yet specialized for next-token prediction. Using text data from OpenWebText (Gokaslan & Cohen, 2019), we train for 100K steps using a batch size of 8192, for a total of approximately 820 million tokens.

## 4.1 SCALING LAWS FOR RECONSTRUCTION ERROR

We first study scaling laws for Switch SAEs, comparing them to dense TopK SAEs at fixed sparsity $k = 32$. In Fig. 1, we show scaling curves of reconstruction MSE error with respect to both FLOPs and number of parameters for Switch SAEs with 8 and 64 experts. We find that Switch SAEs have favorable scaling with respect to FLOPs compared to dense TopK SAEs. In fact, Switch SAEs using ∼1 OOM less compute can often achieve the same reconstruction MSE as TopK SAEs. However, Switch SAEs perform worse at fixed width relative to dense TopK SAEs. Increasing the number of experts improves compute efficiency but reduces parameter efficiency. We hypothesize that this trade-off is a result of features needing to be duplicated across multiple Switch SAE experts, which we discuss in more detail later. Nevertheless, in the right subplot of Fig. 1, we show that the gap between TopK and Switch SAE performance at a fixed width *shrinks* as we scale the number of parameters. Thus, for large-scale experiments, this gap may be imperceptible; since this is the regime in which efficient training is most useful, we believe that this is not a significant weakness of Switch SAEs.

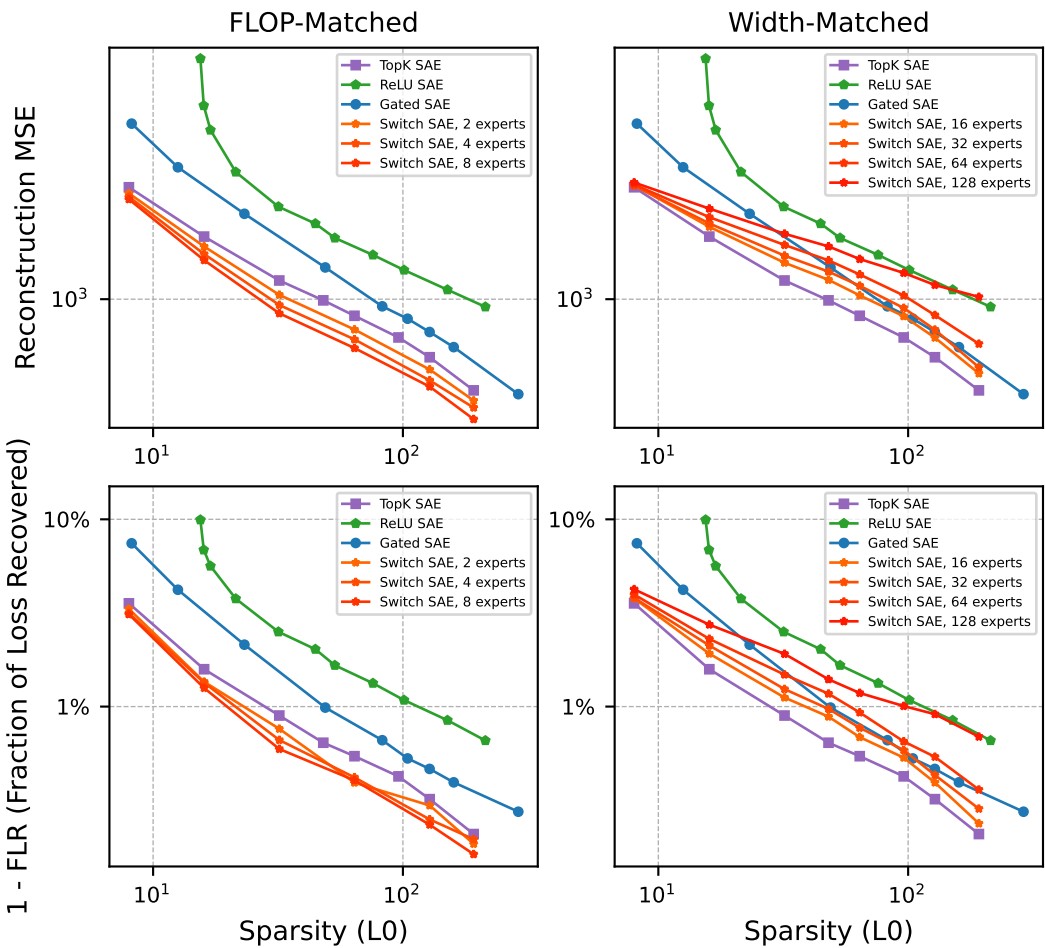

Figure 3: Pareto frontier of sparsity versus (top) reconstruction mean squared error and (bottom) 1 - FLR [fraction of loss recovered]. FLOP-matched Switch SAEs Pareto-dominate TopK SAEs using the same amount of compute (left). Width-matched Switch SAEs perform slightly worse than TopK SAEs but Pareto-dominate ReLU SAEs while performing fewer FLOPs (right).

## 4.2 SPARSITY VS. RECONSTRUCTION ERROR

We now study Switch SAE performance in the reconstruction error vs. sparsity frontier. We benchmark the Switch SAE against the ReLU SAE (Bricken et al., 2023), the Gated SAE (Rajamanoharan et al., 2024a) and the TopK SAE (Gao et al., 2024). We present results for two settings:

1. FLOP-matched: The ReLU, Gated and TopK SAEs are trained with $32 \cdot 768 = 24576$ features. We train Switch SAEs with 2, 4 and 8 experts. Each expert of the Switch SAE with $N$ experts has 24576 features, for a total of $24576 \cdot N$ features. The Switch SAE performs roughly the same number of FLOPs per activation compared to the TopK SAE.

2. Width-matched: Each SAE is trained with $32 \cdot 768 = 24576$ features. We train Switch SAEs with 16, 32, 64 and 128 experts. Each expert of the Switch SAE with $N$ experts has $\frac{24576}{N}$ features. The Switch SAE performs roughly $N$ times fewer FLOPs per activation compared to the TopK SAE.

Note that the router parameters make up a small, insignificant proportion of the total parameters. Across all our experiments, the router parameters make up between $0.002\%$ and $0.3\%$ of the total parameters. For a wide range of sparsity values (L0 between 8 and 128), we report the reconstruction MSE and the fraction of cross-entropy loss recovered (FLR) when the sparse autoencoder output is patched into the language model. A FLR value of 1 corresponds to a perfect reconstruction, while a FLR value of 0 corresponds to a zero-ablation (setting the residual stream to the zero vector).

### 4.2.1 FLOP-MATCHED RESULTS

For the FLOP-matched experiments, we train Switch SAEs with 2, 4 and 8 experts, with the results shown in the left two plots of Fig. 3. The Switch SAEs are a Pareto improvement over the TopK, Gated and ReLU SAEs in terms of both MSE and FLR. As we scale up the number of experts and represent more features, performance continues to increase while keeping computational costs and memory costs (from storing the pre-activations) roughly constant. For a detailed discussion of FLOP calculations, see Appendix A.3.

### 4.2.2 WIDTH-MATCHED RESULTS

For the width-matched experiments, we train Switch SAEs with 16, 32, 64 and 128 experts, with the results shown in the right two plots of Fig. 3. The Switch SAEs consistently underperform compared to the TopK SAE in terms of MSE and FLR, while for the most part outperforming Gated and ReLU SAEs. When L0 is low, Switch SAEs perform particularly well. This suggests that the high frequency features that improve reconstruction fidelity the most for a given activation might lie within the same cluster.

These results demonstrate that Switch SAEs can reduce the number of FLOPs per activation by up to 128x while still retaining the performance of a ReLU SAE. We further believe that Switch SAEs can likely achieve greater acceleration on larger language models.

To show the generality of Switch SAE training, in Appendix B we train on four layers of GPT-2 at residual, attention, and MLP outputs. We also train on a single layer of Gemma 2 2B (Team et al., 2024). We find that Switch SAEs perform well at residual layers on all positions and models tested, but worse on MLP and attention outputs.

## 4.3 FEATURE GEOMETRY

### 4.3.1 FEATURE SIMILARITY

We are interested in why Switch SAEs achieve a worse reconstruction MSE than TopK SAEs of the same size. One hypothesis is that because on any given forward pass only a single expert is activated, some experts will have to learn duplicate features, reducing SAE capacity (this is necessary since there is no perfect split of features into clusters such that features in different clusters never co-occur). We test this hypothesis using a common (see e.g. Braun et al. (2024)) SAE evaluation metric: nearest neighbor cosine similarity.

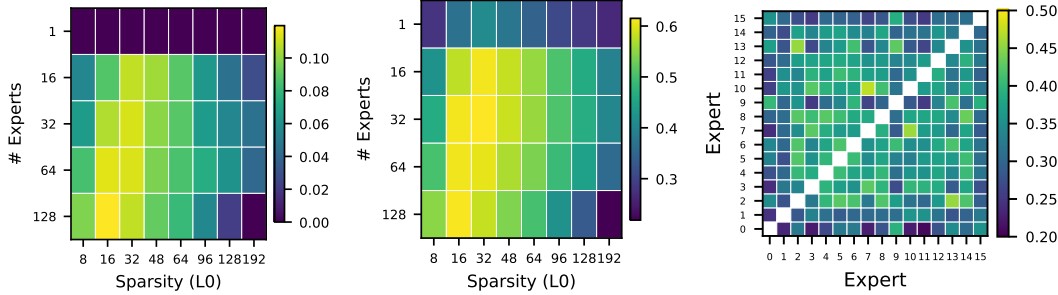

(a) Fraction of SAE decoder vectors with nearest neighbor cosine sim $> 0.9$, width-matched SAEs.

(b) Average cosine sim with nearest neighbor for decoder vectors, width-matched SAEs.

(c) Average cosine sim between nearest neighbors across experts (16 experts, L0 = 64).

Figure 4: Switch SAE feature geometry experiments, measured via cosine similarity between SAE decoder vectors. We find that Switch SAEs with more experts exhibit more feature duplication, but that this effect diminishes for larger L0s. Furthermore, between-expert similarities show that experts specialize moderately; expert 0, for example, has low similarity with most other experts.

One way to use this metric to measure the number of duplicate features in an SAE is to measure the proportion of vectors in the decoder that have a nearest neighbor above a given cosine similarity, since highly similar decoder vectors are likely duplicates. In Fig. 4a, we compare SAEs trained with different numbers of experts and different sparsities on this measure with a threshold of $0.9$, and find that as soon as we train with experts this measure jumpy sharply from a baseline of 0 in TopK SAEs to 5 to 10 percent. In other words, strict duplicates likely reduce Switch SAE capacity by up to $10\%$. This effect is less prevalent at larger L0s. However, we are not just worried about exact duplicates, but also a general shift towards redundancy: in Fig. 4b, we find that a more global metric, cosine similarity of the nearest neighbor averaged across all features, shows a similar pattern across sparsity and number of experts.

Another measure of Switch SAE quality is *expert specialization*, that is, how similar are the sets of features each expert learns. We quantify by averaging the cosine similarity of each feature in expert $A$ with its nearest neighbor in expert $B$. On one end, if the features are perfectly clustered, then each pair of experts should have the same cosine similarity as random blocks of the same size in a TopK SAE of the same size. On the other end, no specialization should result in identical experts. In Fig. 4c, we plot this metric for all pairs of experts in a 16 expert Switch SAE with L0 = 64. Some experts, e.g. expert 0, seem unique, while other pairs of experts, e.g. 10 and 7, are highly similar. All pairs are more similar than random blocks of the same size in a TopK SAE, which has a value of about $0.2$ under this same metric.

### 4.3.2 T-SNE Visualization

To visualize the global structure of SAE features, we show t-SNE projections of the encoder and decoder feature vectors in Fig. 5. We find that encoder features from the same expert cluster together, while decoder features tend to be more diffuse. In the encoder feature t-SNE projection, we can also directly observe feature duplication – around the periphery of the plot we find a variety of isolated points which upon closer inspection are actually tight groupings of multiple features from different experts.

### 4.4 Automated Interpretability

Bricken et al. (2023) evaluate how interpretable sparse autoencoder features using *automated interpretability*: they generate explanations for each feature by giving top activating examples for that feature to a language model, and then ask a language model (in a new context) to predict, using only the description of the feature, on which tokens the described feature fires. More recently, Juang et al. (2024) introduce a more compute efficient method that measures *detection*, whether the explanation allows a language model to predict whether a feature fires at all on a given *context*. Juang et al. (2024) additionally measure detection at each decile of feature activation, as well as on contexts

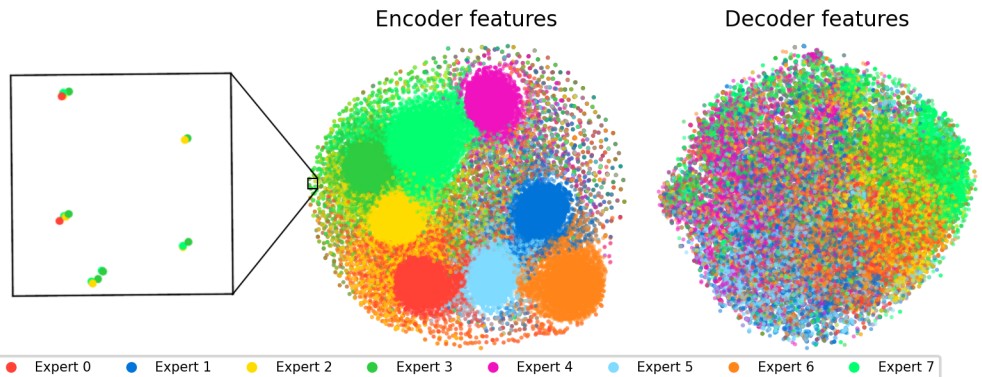

Figure 5: t-SNE projections of encoder and decoder features for a Switch SAE with 65k total features and 8 experts. Encoder features appear to cluster together by expert. Away from these clusters, we see a variety of isolated points which are in fact tight groups of features which have been duplicated across multiple experts. We do not observe as clear clusters in the decoder features.

where the feature does not fire at all (where to be correct the model should report that the feature does not fire). Using the Juang et al. (2024) implementation, we find that FLOP-matched SAE features have similar interpretability as TopK SAEs, but width-matched SAEs have a *higher* true positive rate on contexts where the features fire, while at the same time having a lower true negative rate (Fig. 6). We interpret this result as Switch SAEs being biased towards having duplicate frequent features, which may both be easier to detect and more prone to false positives.

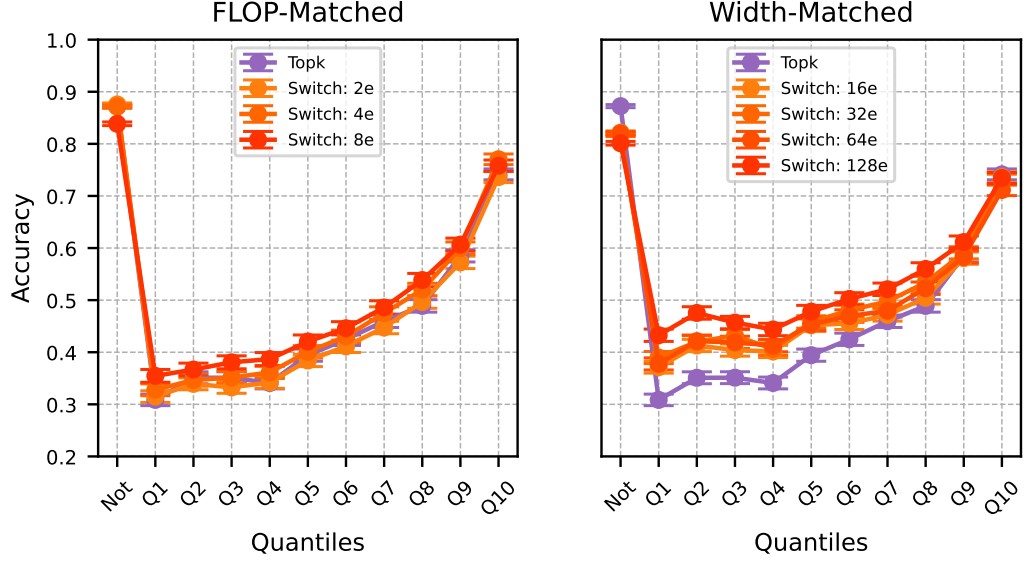

Figure 6: Automated interpretability detection results across SAE feature activation quantiles for 1000 random features, 95% confidence intervals shown. "Not" means that the context comes from a random set of text where the feature was not activating.

## 5    CONCLUSION

Switch SAEs are a promising approach towards scaling sparse autoencoders, as they allow an improvement in the FLOPs vs. MSE scaling law without a significant reduction in feature interpretability. We believe that Switch SAEs may find their best application for huge training runs on large

clusters of GPUs, since in this setting each expert can be split on to a separate GPU, leading to significant wall clock training speed ups. Overall, we believe that this work provides a case study for applying existing machine learning training optimization techniques to the setting of sparse autoencoders for feature extraction from language models; we hope that the investigations in this paper serve as a guide for adapting more such techniques.

**Limitations:** The most significant limitation of the Switch SAE is the reduction in performance of the SAE at a fixed number of parameters (especially for attention and MLP layers, see Appendix B); future work to bridge this gap might investigate feature deduplication techniques, more sophisticated routing architectures, and multiple active experts. Additionally, we investigate only the simplest possible mixture of experts architecture; while this allows us to focus on the question of *whether* and *how* mixture of experts training works at all for sparse autoencoders, it leaves open the question of the maximum performance gain mixture of experts style training might allow. Thus, future work could examine more sophisticated mixture of experts architectures like GShard (Lepikhin et al., 2020), DeepSeekMoE (Dai et al., 2024), and expert choice routing (Zhou et al., 2022) to further optimize performance. Finally, we do not study downstream evaluations of Switch SAEs in this paper; for instance, it is possible that the increased duplication of features between experts complicates feature steering or circuit discovery.

## ACKNOWLEDGMENTS

We used the dictionary learning repository (Marks et al., 2024) to train our SAEs. We would like to thank Samuel Marks and Can Rager for advice on how to use the repository. We would also like to thank Jacob Goldman-Wetzler, Achyuta Rajaram, Michael Pearce, Gitanjali Rao, Satvik Golechha, Kola Ayonrinde, Rupali Bhati, Louis Jaburi, Vedang Lad, Adam Karvonen, Shiva Mudide, Sandy Tanwisuth, JP Rivera and Juan Gil for helpful discussions. This work is supported by Erik Otto, Jaan Tallinn, the Rothberg Family Fund for Cognitive Science, the NSF Graduate Research Fellowship (Grant No. 2141064), IAIFI through NSF grant PHY-2019786, UKRI grant: Turing AI Fellowship EP/W002981/1, and Armasuisse Science+Technology. AM was greatly helped by the MATS program, funded by AI Safety Support.

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

## A  IMPLEMENTATION DETAILS

### A.1  SWITCH SPARSE AUTOENCODER TRAINING DETAILS

We initialize the rows of $\mathbf{W}_{\text{enc}}^i$ to be parallel to the columns of $\mathbf{W}_{\text{dec}}^i$ for all $i$. We initialize both $\mathbf{b}_{\text{pre}}$ and $\mathbf{b}_{\text{router}}$ to the geometric median of a batch of samples, but we do not tie $\mathbf{b}_{\text{pre}}$ and $\mathbf{b}_{\text{router}}$. We additionally normalize the decoder column vectors to unit-norm at initialization and after each gradient step. We remove gradient information parallel to the decoder feature directions to minimize interference with the Adam optimizer. We set the learning rate based on the $\frac{1}{\sqrt{M}}$ scaling law from Gao et al. (2024) and linearly decay the learning rate over the last 20% of training. We do not include neuron resampling (Bricken et al., 2023) or the AuxK loss (Gao et al., 2024), but future work could explore employing these strategies to prevent dead latents in larger models. We optimize $\mathcal{L}_{\text{total}}$ with Adam using the default parameters $\beta_1 = 0.9$, $\beta_2 = 0.999$.

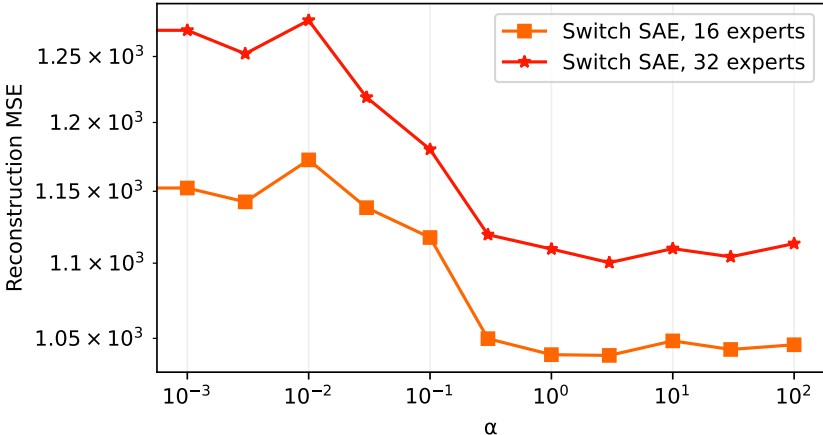

Figure 7: Hyperparameter sweep for $\alpha$. We train fixed-width Switch SAEs with 16 and 32 experts on activations from GPT-2 small. The sparsity level is set to L0=64.

### A.2  HYPERPARAMETER SEARCH

In our objective function, the balance between reconstruction error and the auxiliary loss (which encourages the router to send an equal number of activations to each expert) is controlled by the hyperparameter $\alpha$. We train 16-expert and 32-expert Switch SAEs at a fixed sparsity (L0 = 64) and fixed width (24576 features), but sweep $\alpha$ across the values $[0.001, 0.003, 0.01, 0.03, 0.1, 0.3, 1, 3, 10, 30, 100]$. We find that the reconstruction MSE is not overly sensitive to the value of $\alpha$ and that $\alpha = 3$ performs the best in both the 16-expert and the 32-expert settings (Fig. 7). We default to $\alpha = 3$ for the rest of our experiments.

### A.3  FLOP CALCULATIONS

A single training step of a deep neural network requires roughly $6 \cdot P \cdot D$ FLOPs, where $P$ is the number of active parameters and $D$ is the batch size. Let $d$ be the dimension of the language model activations, $M$ be the total number of features and $N$ be the number of experts. For a TopK SAE, $P = 2Md$. For a Switch SAE, $P = dN + \frac{2Md}{N} \approx \frac{2Md}{N}$ since the router parameters are negligible. Thus, we have

$$\text{FLOPS}_{\text{TopK}}(d, M) = 6 \cdot (2Md) \cdot D,$$

$$\text{FLOPS}_{\text{Switch}}(d, M, N) = 6 \cdot \left( dN + \frac{2Md}{N} \right) \cdot D \approx 6 \cdot \left( \frac{2Md}{N} \right) \cdot D.$$

## B  TRAINING ON ADDITIONAL SITES AND MODELS

In Table 1, we show the results of training on GPT-2 layers $2, 4, 8,$ and $12$ on MLP outputs, attention outputs, and the residual stream. We also train a single SAE on Gemma 2 2B (Team et al., 2024). All SAEs are trained with $k = 64$ with a fixed width; we train the Gemma 2 2B SAEs with width $65536$ and the GPT-2 SAEs with width $24576$. We use $8$ experts for the Switch SAEs. As in the main paper, we find that the decrease in SAE performance for a fixed width minimally effects residual stream training; however, Switch SAEs trained on MLP and attention outputs do suffer a significant performance reduction.

Table 1: Comparison of TopK and Switch SAEs across different models, layers and component types. FVE = Fraction of Variance Explained, FLR = Fraction of Loss Recovered.

| Model | Layer | Type | TopK FVE | Switch FVE | TopK FLR | Switch FLR |
|---|---|---|---|---|---|---|
| GPT-2 | 2 | resid | 0.999 | 0.998 | 0.997 | 0.996 |
| GPT-2 | 2 | attn | 0.919 | 0.884 | 0.997 | 0.968 |
| GPT-2 | 2 | mlp | 0.999 | 0.998 | 0.888 | 0.701 |
| GPT-2 | 4 | resid | 0.997 | 0.997 | 0.996 | 0.995 |
| GPT-2 | 4 | attn | 0.849 | 0.805 | 0.925 | 0.883 |
| GPT-2 | 4 | mlp | 0.877 | 0.819 | 0.853 | 0.777 |
| GPT-2 | 8 | resid | 0.989 | 0.987 | 0.994 | 0.993 |
| GPT-2 | 8 | attn | 0.846 | 0.803 | 0.938 | 0.848 |
| GPT-2 | 8 | mlp | 0.782 | 0.733 | 0.868 | 0.801 |
| GPT-2 | 10 | resid | 0.973 | 0.970 | 0.982 | 0.977 |
| GPT-2 | 10 | attn | 0.892 | 0.863 | 0.942 | 0.883 |
| GPT-2 | 10 | mlp | 0.855 | 0.828 | 0.859 | 0.812 |
| Gemma 2 2B | 12 | resid | 0.961 | 0.955 | 0.991 | 0.989 |

