# OpenReview forum: "Efficient Dictionary Learning with Switch Sparse Autoencoders"
_ICLR.cc/2025/Conference — ICLR 2025 Poster_

### Official Review · Reviewer_x3AE · 2024-11-02

**Soundness:** 3
**Presentation:** 3
**Contribution:** 3
**Rating:** 8
**Confidence:** 4

**Summary:**

This paper introduces **Switch Sparse Autoencoders** (Switch SAEs). Switch SAEs are a new SAE variant, which is a popular area of research in interpretability. Switch SAEs are more efficiently to train and run at inference time for similar reasons to why Mixture-of-Expert (MoE) LLMs are more efficient than dense LLMs.

This paper claims that, fixing the number of FLOPs, Switch SAEs pareto dominate several leading SAE variants. Fixing the total number of features SAEs have however, Switch SAEs seem less performant. The paper then shows Switch SAEs have as interpretable metrics as dense SAEs as measured by an automated metric. There are some surface level studies into feature duplication, suggesting that this sometimes happens. All experiments are on GPT-2 Small's residual stream at layer 8.

Here are my references for all sections in this review:

[1] Switch MoE: https://arxiv.org/abs/2101.03961

[2] DeepSeek MoE: https://arxiv.org/abs/2401.06066

[3] GShard: https://arxiv.org/abs/2006.16668

[4] Expert-choice; https://arxiv.org/abs/2202.09368

[5] Ghost Grads: https://transformer-circuits.pub/2024/jan-update#dict-learning-resampling

[6] Updated Anthropic training method: https://transformer-circuits.pub/2024/april-update/index.html#training-saes

[7] Gated SAEs: https://arxiv.org/abs/2404.16014

[8] JumpReLU SAEs: https://arxiv.org/abs/2407.14435

[9] Anthropic Set Of Metrics Where JumpReLU does not reproduce: https://transformer-circuits.pub/2024/august-
update/index.html#interp-evals

[10] Scaling Monosemanticity: https://transformer-circuits.pub/2024/scaling-monosemanticity/

[11] SFC: https://arxiv.org/abs/2403.19647

**Strengths:**

1. **This paper addresses an important and novel direction that may be impactful for future work.** There are already several very large training runs that have trained large SAEs, and a key bottleneck is the autoencoder training (getting and potentially storing billions of activations from frontier models is also a bottleneck, however). The paper provides promising evidence for the MoE SAE approach.

2. **The evaluation is clear and relatively comprehensive**. Addressing FLOP-matched results and feature interpretability compared to baselines is very clear.

3. **The authors do sensible sanity checks of the approach.** Duplicated features could be a serious issue, and the analysis of how frequent they were showed they weren't an enormous proportion of features, but still present. I like the transparency of this section.

**Weaknesses:**

1. **There is no application of the Switch SAE in the paper.** Recently, SAEs have been applied to steering [10] and circuit finding [11] as well as many other applications in AI safety and capabilities. These applications rely on a sparse set of features explaining model behavior. But if there are N sets of sparse features, different sets for each of the experts, then these applications may not want to use Switch SAEs despite performance benefits. This is a deeper problem than merely "there may be duplicated features in different experts (as measured by cosine sim / t-SNE)". For example, on a given dataset of interest, such as a dataset used to generate an SFC graph [11] or the set of prompts related to e.g. the Golden Gate Bridge [10], there may be different ways the different experts reconstruct similar prompts. As a toy example, on a set of prompts where the "Queen" concept is always present, on one prompt Expert 1 may be chosen, and encode `V(Queen) = Feature(Woman) + Feature(Royalty)`. But on another prompt Expert 2 may be chosen and have a dedicated feature `Feature(Queen)`. Let me know if I need to make this explanation clearer.

2. **This paper bases their approach off of the Switch MoE architecture [1], but this is a weak MoE variant**. My understanding is that the switch MoE architecture is weaker than different MoE architectures. For example [2] benchmarks Switch MoEs against GShard [3] and their own DeepSeekMoE [2], and find that Switch MoEs are weakest.

Additionally, this paper requires an auxiliary loss to balance MoE load. While this is popular in conventional MoE work [1, 2, 3], other works propose to do this [4]. In the context of SAEs, several improvements have involved dropping auxiliary loss terms (e.g. [4] was replaced by [5] by the Anthropic group, and [7] was replaced by [8] by the Google DeepMind group). So I am also concerned that auxiliary loss terms are worse than (e.g.) expert-choice variants of MoEs in training MoE SAEs.

I would like to hear why these design choices were made, and think the paper should be edited if accepted to the conference to discuss these variants for future work, and to hedge the conclusions more.

3. **This paper only uses one Language Model**. GPT-2 Small has a specfic architecture which is unlike modern LLMs in several ways (only uses GeLU not gated MLPs, uses absolute positional embeddings, ...) which may mean the results do not generalize. (But this is unlikely since most work has found so far that SAE approaches that work on model 1 also work on model 2).

4. **The benefits of MoE SAEs are only at very large scale, and GPT-2 Small 800M token training runs are not large scale**. When scaling ML techniques it is often found that small-scale hacks perform worse in large setups. This is still a plausible issue given the results in paper.

Note that, despite how I found more strengths than weaknesses, these weaknesses are more technical and subtle. I think the paper overall is strong, and it required harder thinking to realise some issues with it.

**Questions:**

1. How can I calculate the FLOPs for a Switch SAE with given num experts, width, input dim and training datapoints? I did not see this in the paper, and therefore sentences like "As we scale up the number of
experts and represent more features, performance continues to increase while keeping computational
costs and memory costs (from storing the pre-activations) roughly constant" are very confusing.

(My GUESS at what that sentence is gesturing at is that fixing Switch SAE width, increasing number of experts decreases the encoder forward pass FLOP but not the backward pass FLOP. And after thinking about it for a minute, I think I see how increasing number of experts doesn't increase FLOP so long as you only route acts to 1 expert, always. But this was a non-trivial inference and I think it should be in the paper)

2. Are the authors able to open source training code and/or weights when this is deanonymized? Since there have been known cases in the SAE literature where promising work has not replicated [8, 9], open weights and training code will enable downstream validation of this technique to be much stronger.

3. Do you have any experiments on different models or different sites?

---

> ### Author Response · Authors · 2024-11-28
> **Response, Part 1**
>
> Thank you for your extremely thoughtful comments. We greatly appreciate you taking the time to think deeply about our work.
>
> **Response to weaknesses, part 1**
> > There is no application of the Switch SAE in the paper. Recently, SAEs have been applied to steering [10] and circuit finding [11] as well as many other applications in AI safety and capabilities. These applications rely on a sparse set of features explaining model behavior. But if there are N sets of sparse features, different sets for each of the experts, then these applications may not want to use Switch SAEs despite performance benefits. This is a deeper problem than merely "there may be duplicated features in different experts (as measured by cosine sim / t-SNE)". For example, on a given dataset of interest, such as a dataset used to generate an SFC graph [11] or the set of prompts related to e.g. the Golden Gate Bridge [10], there may be different ways the different experts reconstruct similar prompts. As a toy example, on a set of prompts where the "Queen" concept is always present, on one prompt Expert 1 may be chosen, and encode V(Queen) = Feature(Woman) + Feature(Royalty). But on another prompt Expert 2 may be chosen and have a dedicated feature Feature(Queen). Let me know if I need to make this explanation clearer.
>
> This is an excellent and thoughtful point that we had not considered, thank you! We have added a discussion of it to the limitations section of our conclusion. Interestingly, however, we believe that our existing experiments show that it may not actually be a large issue. Specifically, our auto-interp evaluation asks an LLM to predict when a feature fires depending on input text. If your concern was happening frequently, then detection scores should be much lower, since sometimes the “same” feature would fire on similar types of text in expert 1 versus expert 2 in different contexts, and we would not be able to predict that “cross-expert feature” successfully. However, detection scores are only slightly lower for the Switch SAE. Thus, we believe that reconstruction MSE, loss recovered and auto-interpretability score are together a good proxy for downstream tasks, even in the Switch SAE setting.
>
> > This paper bases their approach off of the Switch MoE architecture [1], but this is a weak MoE variant. My understanding is that the switch MoE architecture is weaker than different MoE architectures. For example [2] benchmarks Switch MoEs against GShard [3] and their own DeepSeekMoE [2], and find that Switch MoEs are weakest.
>
> Thank you for pointing this out, we agree that other mixture of expert variants may perform better. However, we do not view this as solely a weakness of our work; indeed, using the simplest possible MoE variant in the paper allows us to focus on whether and how the technique works at all for language model feature learning, and thus provides a solid baseline that future work can improve upon. We have added a section to our conclusion discussing this future work, thank you again!
>
> > Additionally, this paper requires an auxiliary loss to balance MoE load. While this is popular in conventional MoE work [1, 2, 3], other works propose to do this [4]. In the context of SAEs, several improvements have involved dropping auxiliary loss terms (e.g. [4] was replaced by [5] by the Anthropic group, and [7] was replaced by [8] by the Google DeepMind group). So I am also concerned that auxiliary loss terms are worse than (e.g.) expert-choice variants of MoEs in training MoE SAEs.
>
> This is another excellent point, thank you! We agree that an auxiliary loss term is less elegant, but for the same reasons as above we leave exploring additional mixture of expert architectures to future work. Also, as shown in the new Figure 7 in the appendix, Switch SAEs at least seem robust to the auxiliary weighting term $\alpha$, so this is not a sensitive parameter that needs to be carefully optimized.

---

> ### Author Response · Authors · 2024-11-28
> **Response, Part 2**
>
> **Response to weaknesses, part 2**
>
> > This paper only uses one Language Model. GPT-2 Small has a specfic architecture which is unlike modern LLMs in several ways (only uses GeLU not gated MLPs, uses absolute positional embeddings, ...) which may mean the results do not generalize. (But this is unlikely since most work has found so far that SAE approaches that work on model 1 also work on model 2). The benefits of MoE SAEs are only at very large scale, and GPT-2 Small 800M token training runs are not large scale. When scaling ML techniques it is often found that small-scale hacks perform worse in large setups. This is still a plausible issue given the results in paper.
>
> Thank you for pointing these concerns out! We have added the following table in the appendix, which shows results for training GPT-2 TopK and Switch SAEs at the same width and sparsity on additional layers and sites, as well as Gemma-2B SAEs on a single layer, and have added a reference to it in section 4.2.2:
>
>
> | Model   |   Layer | Type   |   TopK FVE |   Switch FVE |   TopK FR |   Switch FR |
> |:--------|--------:|:-------|-----------:|-------------:|----------:|------------:|
> | GPT-2   |       2 | resid  |      0.999 |        0.998 |     0.997 |       0.996 |
> | GPT-2   |       2 | attn   |      0.919 |        0.884 |     0.997 |       0.968 |
> | GPT-2   |       2 | mlp    |      0.999 |        0.998 |     0.888 |       0.701 |
> | GPT-2   |       4 | resid  |      0.997 |        0.997 |     0.996 |       0.995 |
> | GPT-2   |       4 | attn   |      0.849 |        0.805 |     0.925 |       0.883 |
> | GPT-2   |       4 | mlp    |      0.877 |        0.819 |     0.853 |       0.777 |
> | GPT-2   |       8 | resid  |      0.989 |        0.987 |     0.994 |       0.993 |
> | GPT-2   |       8 | attn   |      0.846 |        0.803 |     0.938 |       0.848 |
> | GPT-2   |       8 | mlp    |      0.782 |        0.733 |     0.868 |       0.801 |
> | GPT-2   |      10 | resid  |      0.973 |        0.97  |     0.982 |       0.977 |
> | GPT-2   |      10 | attn   |      0.892 |        0.863 |     0.942 |       0.883 |
> | GPT-2   |      10 | mlp    |      0.855 |        0.828 |     0.859 |       0.812 |
> | Gemma*   |      12 | resid  |      0.941 |        0.931 |     0.979 |       0.969 |
>
>
> The * means that these SAEs have not yet trained to convergence, but should converge in the next couple of days. Once they are done we plan to update the table in this comment and add a reply.
>
> Overall, we find that our results replicate on a larger model, Gemma 2 2B, and additional GPT-2 residual layers, but do not replicate as well on earlier attention and MLP layers of GPT-2. Currently, residual stream SAEs are the most widely used by the community, so we believe this result does not strongly negatively affect our contribution, but we certainly are interested in future work investigating this disparity.  We have added a discussion of this limitation to our conclusion.
>
> We further expect mixture of experts based techniques to work well at scale for two reasons:
>
> - We see increased speedup with larger SAEs in Figure 1
>
> - Mixture of expert models have been shown to work very well at large scales in language models, see e.g. Mixtral of Experts [1]
>
> **Response to questions**
>
> > How can I calculate the FLOPs for a Switch SAE with given num experts, width, input dim and training datapoints? I did not see this in the paper, and therefore sentences like "As we scale up the number of experts and represent more features, performance continues to increase while keeping computational costs and memory costs (from storing the pre-activations) roughly constant" are very confusing.
>
> Thank you very much for bringing up this concern. We have added Appendix A.3 to explain the FLOP calculations and have added a reference to Appendix A.3 in section 4.2.1. We apologize for not including this initially and greatly appreciate your attention to detail.
>
> > Are the authors able to open source training code and/or weights when this is deanonymized? Since there have been known cases in the SAE literature where promising work has not replicated [8, 9], open weights and training code will enable downstream validation of this technique to be much stronger.
>
> Yes, we will open source our training code and weights when our work is de-anonymized. We are also planning on integrating Switch SAEs within existing tools such as SAE Lens (https://jbloomaus.github.io/SAELens/).
>
> > Do you have any experiments on different models or different sites?
>
> Yes, see above, thank you!
>
> If our comments resolve your concerns, we would appreciate it if you would consider raising your score!
>
> [1] https://mistral.ai/news/mixtral-of-experts/

---

> ### Comment · Reviewer_x3AE · 2024-11-28
> **Thanks: I am raising my score from 6 to 8**
>
> Thanks for noting all my points as limitations in the paper, or doing the additional training runs / experiments to address some doubts. It is strange that attn and MLP sites have worse switch SAE performance, but they are less important than the residual stream site, so I'm confident this paper should be accepted.

---

> > ### Author Response · Authors · 2024-12-01
> >
> > Thank you! Here are the final numbers of the converged Gemma-2 2B SAEs:
> >
> > | Model   |   Layer | Type   |   TopK FVE |   Switch FVE |   TopK FR |   Switch FR |
> > |:--------|--------:|:-------|-----------:|-------------:|----------:|------------:|
> > | Gemma   |      12 | resid  |      0.961 |        0.955 |     0.991 |       0.989 |

---

### Official Review · Reviewer_g9rJ · 2024-11-04

**Soundness:** 2
**Presentation:** 2
**Contribution:** 2
**Rating:** 6
**Confidence:** 3

**Summary:**

This paper proposes an extension to Sparse Autoencoders (SAEs) in the form of switch SAEs. Switch SAEs reduce the memory consumption of storing the sparse activations by splitting the representation space across multiple SAE experts, of which only one is used at a time. Experiments show that switch SAEs achieve better reconstructions with the same number of FLOPs as current methods at the cost parameter efficency. The paper also explores the features learned by the experts and observe that some features cluster within experts and others are duplicated across them.

**Strengths:**

The strengths of the paper are its simple approach and side studies.
- The approach enables a trade-off between model parameters and computational requirements which should be useful to practitioners.
- The auxiliary experiments are well-motivated and support the author’s hypotheses. Specifically, the discussions of redundant representations accompanied by t-SNE visualizations are illuminating.
- The limitations (parameter efficiency and feature duplication) are discussed in an informative way.

**Weaknesses:**

The main weakness of the paper is its quantitative evaluation.
- The hyperparameter $\alpha$ is set to 3 (line 258) without further discussion. How was this value chosen? Is this a reasonable default in most settings or is tunning required based on the model and dataset?
- The abstract claims a “substantial Pareto improvement” for a fixed training budget. This corresponds to the FLOP-matched experiments shown in the bottom left plot of Figure 3. It is not directly obvious that this constitutes a substantial improvement. The difference looks small enough to warrant repeated experiments to ensure its significance. A table of improvements with confidence intervals would make the difference much clearer.
- Figure 4 could use an explanatory caption.

**Questions:**

- Why are the proportion of activations routed and the routing probability used in the auxiliary loss? Intuitively, both convey the same information, with one being the distribution and the other samples from it.
- A batch size of 8192 is used during training (line 269). Is a large batch size required for stable training of switch sparse autoencoders, and does this differ from the baseline methods?
- Finally a broader question about the utility of scaling SAEs. Do the learned features need to be manually interpreted? If so, how is increasing the number of parameters a useful approach to interpretability if it also increases the required manual annotation?
- The references seem biased towards recent years. Besides the Adam optimizer and GPU acceleration, no references exist from before 2016, despite sparse autoencoders, sparsity, topK, and MoEs being popular research topics much earlier than that. Are these not an important background to sparse SAEs?

**Minor suggestions that do not individually affect the score**
- Line 295: “For a wide range of sparsity (L0) values”. Specify the range.
- Line 298: Clarify what is meant by a zero-ablation.
- Line 317: “Switch SAEs can likely achieve greater acceleration on larger language models”. Expand on this.

---

> ### Author Response · Authors · 2024-11-28
> **Response Part 1**
>
> Thank you for your thoughtful comments. We greatly appreciate you taking the time to review our work.
>
> **Response to weaknesses**
> > The hyperparameter $\alpha$ is set to 3 (line 258) without further discussion. How was this value chosen? Is this a reasonable default in most settings or is tunning required based on the model and dataset?
>
> Thank you for bringing this to our attention. We chose $\alpha = 3$ based on a preliminary hyperparameter sweep, where we found that results were not very sensitive to $\alpha$ and that $\alpha = 3$ was a reasonable default. We trained 16-expert and 32-expert Switch SAEs at a fixed sparsity (L0 = 64) and width (24576 features) and swept $\alpha$ across the values [0.001, 0.003, 0.01, 0.03, 0.1, 0.3, 1, 3, 10, 30, 100]. We have added a plot (Figure 7) corresponding to this hyperparameter sweep to Appendix A.2. We apologize for not including this in our initial submission.
>
> > The abstract claims a “substantial Pareto improvement” for a fixed training budget. This corresponds to the FLOP-matched experiments shown in the bottom left plot of Figure 3. It is not directly obvious that this constitutes a substantial improvement. The difference looks small enough to warrant repeated experiments to ensure its significance. A table of improvements with confidence intervals would make the difference much clearer.
>
> The substantial Pareto improvement is much more obvious in the top left plot of Figure 3, showing reconstruction MSE. The substantial Pareto improvement for a fixed training budget is also apparent in our scaling laws (specifically, the left subplot of Figure 1). The Pareto improvement is less visually obvious in the bottom left plot of Figure 3 because the fraction of loss recovered (FLR) metric is saturated. We have inverted the y-axis (plotting 1 - FLR as opposed to FLR) in our updated submission to highlight the difference more clearly. In general, SAEs trained to convergence are very stable in terms of their reconstruction MSE and FLR.
>
> > Figure 4 could use an explanatory caption.
>
> Thank you for bringing this to our attention. We have added an explanatory caption to our submission. The caption now reads “Switch SAE feature geometry experiments, measured via cosine similarity between SAE decoder vectors. We find that Switch SAEs with more experts exhibit more feature duplication, but that this effect diminishes for larger L0s. Furthermore, between-expert similarities show that experts specialize moderately; expert 0, for example, has low similarity with most other experts.”
>
>
> **Response to questions, part 1**
> > Why are the proportion of activations routed and the routing probability used in the auxiliary loss? Intuitively, both convey the same information, with one being the distribution and the other samples from it.
>
> Thank you for this excellent question. The purpose of the auxiliary loss is to encourage the model to send an equal number of activations to each expert in order to reduce overhead. Thus, we aim to drive the f-vector towards the uniform distribution. However, we cannot directly optimize this objective because the f-vector is not differentiable. So, we instead optimize the scaled dot-product between the f-vector and the P-vector, which is feasible because the P-vector is differentiable. Note that this auxiliary loss is not original to our work; it comes from [1].
>
> > A batch size of 8192 is used during training (line 269). Is a large batch size required for stable training of switch sparse autoencoders, and does this differ from the baseline methods?
>
> Thank you for asking this question. A batch size of 8192 is typical for training sparse autoencoders. [2] similarly uses a batch size of 8192, [3] and [4] use a batch size of 4096 and [5] uses a much greater batch size of 131072.
>
> > Finally a broader question about the utility of scaling SAEs. Do the learned features need to be manually interpreted? If so, how is increasing the number of parameters a useful approach to interpretability if it also increases the required manual annotation?
>
> Thank you for expressing this concern. In the right subplot of Fig. 1, we demonstrate that the gap between TopK and Switch SAE performance at a fixed width *shrinks* as we scale the number of parameters. For this reason, we expect that Switch SAEs will require a very small amount of additional features to achieve the same reconstruction loss on frontier language models.
>
> In addition, there exists a separate line of work dedicated to efficient automatic interpretability. For example, recent work [6] open-sourced an automated pipeline for efficiently interpreting millions of features.

---

> ### Author Response · Authors · 2024-11-28
> **Response Part 2**
>
> **Response to questions, part 2**
> > The references seem biased towards recent years. Besides the Adam optimizer and GPU acceleration, no references exist from before 2016, despite sparse autoencoders, sparsity, topK, and MoEs being popular research topics much earlier than that. Are these not an important background to sparse SAEs?
>
> Thank you for bringing this to our attention. We have added additional citations to our submission to address this. For sparse autoencoders, we added [7], [8], [9] to section 1 and section 2.3. For sparsity, we added [10], [11], [12], [13] to section 2.3. For TopK, we added [14] to section 1 and section 2.3. For MoE, we added [15], [16], [17], [18], [19], [20] to section 2.1.
>
> **Response to suggestions**
>
> > Line 295: “For a wide range of sparsity (L0) values”. Specify the range.
>
> Thank you for pointing this out! The range is 8 - 128; we have added this clarification to our submission.
>
> > Line 298: Clarify what is meant by a zero-ablation.
>
> Thank you also for mentioning this! Zero-ablation corresponds to setting the residual stream to all 0s at that layer; we have now added this clarification to our submission. Note that this is a standard metric used for evaluating SAEs, see e.g. [2], [4], [21].
>
> > Line 317: “Switch SAEs can likely achieve greater acceleration on larger language models”. Expand on this.
>
> We believe this is true for two main reasons. First, we see an increased speedup with larger SAEs in Figure 1. Second, mixture of expert models have been shown to work very well at large scales in language models, see e.g., Mixtral of Experts [22].
>
> If our comments resolve your concerns, we would appreciate it if you would consider raising your score!
>
> [1] Switch Transformers: Scaling to Trillion Parameter Models with Simple and Efficient Sparsity, Fedus et al.
>
> [2] Towards monosemanticity: Decomposing language models with dictionary learning, Bricken et al.
>
> [3] Update on how we train SAEs, Conerly et al.
>
> [4] Improving dictionary learning with gated sparse autoencoders, Rajamanoharan et al.
>
> [5] Scaling and evaluating sparse autoencoders, Gao et al.
>
> [6] Automatically Interpreting Millions of Features in Large Language Models, Paulo et al.
>
> [7] Sparse deep belief net model for visual area v2, Lee et al.
>
> [8] Building high-level features using large scale unsupervised learning, Le et al.
>
> [9] Zero-bias autoencoders and the benefits of co-adapting features, Konda et al.
>
> [10] Sparse coding with an overcomplete basis set: A strategy employed by v1?, Olshausen et al.
>
> [11] Sparse and redundant representations: from theory to applications in signal and
> image processing, Elad.
>
> [12] Method of optimal directions for frame design, Engan et al.
>
> [13] K-svd: An algorithm for designing overcomplete dictionaries for sparse representation, Aharon et al.
> [14] K-sparse autoencoders, Makhzani & Frey.
>
> [15] Adaptive mixtures of local experts, Jacobs et al.
>
> [16] Hierarchical mixtures of experts and the em algorithm, Jordan & Jacobs.
>
> [17] Mixtures of gaussian processes, Tresp.
>
> [18] A parallel mixture of svms for very large scale problems, Collobert et al.
>
> [19] Infinite mixtures of gaussian process experts, Rasmussen & Ghahramani.
>
> [20] Expert gate: Lifelong learning with a network of experts, Aljundi et al.
>
> [21] Open Source Sparse Autoencoders for all Residual Stream Layers of GPT2-Small, Bloom.
>
> [22] https://mistral.ai/news/mixtral-of-experts/.

---

> > ### Comment · Reviewer_g9rJ · 2024-12-01
> > **Raised score and additional questions**
> >
> > Thank you for the response. The answers clarifying the model's design ($\alpha$, auxiliary loss, and batch size), and experimental setting (range of sparsity values, meaning of zero-ablation) are helpful. Including the explanation of the auxiliary loss' differentiability in Section 3.2 and explicitly referring unfamiliar readers to Fedus et al. would be useful. The extended related works section is an appreciated improvement in this regard.
> >
> > Based on this response, those to the other reviewer's questions, and the revised paper, I will raise my score to 6 and lean towards accepting the paper. As mentioned in my initial review, the paper's main weakness is its quantitative evaluation. The results are somewhat limited: single-run, mostly GPT2 (as pointed out by Reviewer x3AE, the addition of Gemma 2 2B in Appendix B is welcome), so how far the results extend is not entirely clear yet. At the same time, the proposed approach is simple and likely useful to practitioners looking to trade off parameters and FLOP efficiency.
> >
> > **Additional questions**
> >
> > - On line 479, you mention potential *wall clock time* improvements. Throughout the paper, the main point of comparison is FLOPs. Did you observe improvements to wall clock time? What setting were the experiments run in (mulit-GPU, multi-node?), and how did the wall clock times compare to the theoretical FLOP calculations? Considering that practical speedups are one of this work's main appeals, I think this is valuable to include.
> > - I still think the main results should be presented in tables to complement the figures to make the exact results clear. Among other things, this makes comparing results in reproductions or follow-up work to the results in this work. Reading off approximate numbers from the graph should not be necessary. I appreciate the table in Appendix B (on the flip side these could also be accompanied by a figure!). Could a similar table be added for the main results?

---

> > > ### Author Response · Authors · 2024-12-02
> > > **Response**
> > >
> > > We are happy to hear that our response was helpful. We will add the explanation of the auxiliary loss to Section 3.2 in a future revision. Thank you for your detailed feedback, which has greatly improved the quality of our paper.
> > >
> > > **Response to additional questions**
> > > > On line 479, you mention potential wall clock time improvements. Throughout the paper, the main point of comparison is FLOPs. Did you observe improvements to wall clock time? What setting were the experiments run in (mulit-GPU, multi-node?), and how did the wall clock times compare to the theoretical FLOP calculations? Considering that practical speedups are one of this work's main appeals, I think this is valuable to include.
> > >
> > > We ran these experiments in a single-GPU setting, so we were not able to fully realize the wall clock time improvements. [1] describes an effective strategy for expert and data parallelism, but we leave incorporating this to future work. As we mention in our conclusion, we believe that Switch SAEs may find their best application for huge training runs on large clusters of GPUs, since in this setting each expert can be split on to a separate GPU, leading to significant wall clock training speed ups.
> > >
> > > > I still think the main results should be presented in tables to complement the figures to make the exact results clear. Among other things, this makes comparing results in reproductions or follow-up work to the results in this work. Reading off approximate numbers from the graph should not be necessary. I appreciate the table in Appendix B (on the flip side these could also be accompanied by a figure!). Could a similar table be added for the main results?
> > >
> > > Thank you for the suggestion. We are happy to add this table to the appendix in a future revision.
> > >
> > > [1] Switch Transformers: Scaling to Trillion Parameter Models with Simple and Efficient Sparsity, Fedus et al.

---

### Official Review · Reviewer_uAzQ · 2024-11-08

**Soundness:** 3
**Presentation:** 3
**Contribution:** 3
**Rating:** 8
**Confidence:** 4

**Summary:**

The paper introduces switch SAE's which is a combination of switch layers with top-k SAEs. Instead of decomposing a latent directly it is forwarded to one of multiple SAE encoder decoder pairs. The authors show that given a FLOP budget the switch SAE reaches a lower reconstruction loss faster, while for a fixed number of parameters the vanilla top-k SAEs achieve lower reconstruction loss. That is for a low reconstruction loss with switch SAEs more features in total are required (but they are cheaper to train). Switch SAE features are shown to also be interpretable.

**Strengths:**

The authors explore a combination of methods that makes a lot of sense (switch layers and top-k SAEs) and analyze the resulting method in a rigorous and convincing way. They make the interesting findings that switch SAE's are in terms of FLOPS actually cheaper to train (for a given target reconstruction loss) and at the same time require more features in total (which will be an additional challenge for the usual subsequent automatic interpretation of the learnt features). Also they show that the switch SAE features are interpretable.

**Weaknesses:**

The idea maybe can be considered as a little bit incremental, but given the very good execution of the paper I don't think that's a big problem.

The switch SAEs require more features for the same reconstruction loss, which down the road: for automatic interpretability and SAE feature based interventions presents significant drawbacks. The cost of automatic annotation scales linearly with the number of features to annotate and in my opinion having a giant number of SAE features also makes them less human-compatible/interpretable.

**Questions:**

Do you think it is possible to set up switch SAE in a way that each SAE becomes a domain expert? E.g., there could be one SAE per language or something like that.

---

> ### Author Response · Authors · 2024-11-28
> **Response**
>
> Thank you for your thoughtful comments. We are glad that you enjoyed our paper!
>
> **Response to weaknesses**
>
> > The switch SAEs require more features for the same reconstruction loss, which down the road: for automatic interpretability and SAE feature based interventions presents significant drawbacks. The cost of automatic annotation scales linearly with the number of features to annotate and in my opinion having a giant number of SAE features also makes them less human-compatible/interpretable.
>
> Thank you for expressing this concern. In the right subplot of Fig. 1, we demonstrate that the gap between TopK and Switch SAE performance at a fixed width *shrinks* as we scale the number of parameters at a fixed sparsity. For this reason, we expect that Switch SAEs will require a very small amount of additional features to achieve the same reconstruction loss for the very wide SAEs trained on frontier language models.
>
> In addition, there exists a separate line of work dedicated to efficient automatic interpretability. For example, recent work [1] open-sourced an automated pipeline for efficiently interpreting millions of features.
>
> **Response to questions**
> > Do you think it is possible to set up switch SAE in a way that each SAE becomes a domain expert? E.g., there could be one SAE per language or something like that.
>
> Thank you for asking this question. Typically, mixture-of-expert models are not cleanly interpretable (e.g., there isn’t a “chemistry” expert versus a “physics” expert). It may be possible to enforce that each SAE focuses on a different domain via a different router architecture, but in this work we focus on efficiency and feature quality. Other recent works, e.g. [2], [3], [4], also examine post-hoc clustering of SAE features; each cluster can then be seen as a domain expert.
>
> [1] Automatically Interpreting Millions of Features in Large Language Models, Paulo et al.
>
> [2] Not all language model features are linear, Engels et al.
>
> [3] The Geometry of Concepts: Sparse Autoencoder Feature Structure, Li et al.
>
> [4] HDBSCAN is Surprisingly Effective at Finding Interpretable Clusters of the SAE Decoder Matrix, Lim et al.

---

### Official Review · Reviewer_GyB2 · 2024-11-09

**Soundness:** 3
**Presentation:** 3
**Contribution:** 2
**Rating:** 6
**Confidence:** 3

**Summary:**

This paper introduces Switch sparse autoencoders for learning sparse representations of neural network activations. Switch SAEs include a mixture of TopK SAEs and a router to ensure that only one expert processes each input.

**Strengths:**

This work presents an idea to apply a mixture-of-experts layer to sparse autoencoder training and propose Switch sparse autoencoders. However, the architecture itself and the training technique are straightforward applications of existing work.

The author conducts experiments comparing the Pareto frontiers of proposed Switch SAEs against a single SAE baseline, showing their better reconstruction performance at similar sparsity levels. I think it is somewhat expected for the mixture model due to expert specialization and the additional router parameters. Moever, the training of this architecture is more complex and costly.

The worse performance on the same width setting is indeed a drawback of this work, which might limit its applications.

**Weaknesses:**

See above.

**Questions:**

Even though only one expert is active, I wonder what sparsity patterns look like for other experts and how much they overlap. Would it make sense to compare using global sparsity?

---

> ### Author Response · Authors · 2024-11-28
> **Response**
>
> Thank you for your thoughtful comments. We greatly appreciate you taking the time to review our work.
>
> **Response to weaknesses**
> > However, the architecture itself and the training technique are straightforward applications of existing work.
>
> We agree that mixture-of-experts and sparse autoencoders are well established techniques in the literature. However, this work is the first to demonstrate that these ideas can be synergistically combined to enable more efficient large-scale dictionary learning on model activations. We believe our contributions are important in that they show not only scaling laws and performance tradeoffs, but also examine more subtle questions of feature duplication, geometry and interpretability.
>
> > The author conducts experiments comparing the Pareto frontiers of proposed Switch SAEs against a single SAE baseline, showing their better reconstruction performance at similar sparsity levels.
>
> As a small note, in Fig. 3, we compare against three separate SAE baselines (ReLU SAE [1], Gated SAE [2] and TopK SAE [3]), not just one.
>
> > I think it is somewhat expected for the mixture model due to expert specialization and the additional router parameters.
>
> Thank you for expressing this concern. The router parameters make up a very small proportion of the overall parameters. Across all our experiments, the router parameters make up between **0.002% and 0.3% of the total parameters**. We apologize for not making this more clear and have added additional text to our submission to help clarify this. Thus, expert specialization is indeed the reason the Switch SAE performs better, but we do not think this was necessarily expected; it depends on how clusterable the model’s feature manifold is, and one of the main findings of this work (see e.g. Figure 5) is that empirically it is clusterable!
>
> > Moever, the training of this architecture is more complex and costly.
>
> Thank you for bringing up this concern. In the left subplot of Fig. 1, we demonstrate that training a Switch SAE is approximately *10x less costly* than training a TopK SAE with the same reconstruction error. The training procedure is also only slightly more complex, and thus we believe it will be well worth the associated efficiency gains in many cases. We also plan to release the non-anonymized link to our code once we are allowed to, which should make future re-implementations in modern SAE training libraries significantly easier.
>
> > The worse performance on the same width setting is indeed a drawback of this work, which might limit its applications.
>
> We agree with you that this is a limitation of our current work. However, as shown in the right subplot of Fig. 1, the gap between TopK and Switch SAE performance at a fixed width *shrinks* as we scale the number of parameters. Thus, we expect that this drawback will be imperceptible at large scales and will not limit its applications. We apologize for not making this more clear and have added additional text to our submission to help clarify this.
>
> In addition, we believe that future work might further decrease the gap in the same-width setting. In our conclusion, we cite “feature deduplication techniques, more sophisticated routing architectures, and multiple active experts” as possible directions.
>
> **Response to questions**
> > Even though only one expert is active, I wonder what sparsity patterns look like for other experts and how much they overlap. Would it make sense to compare using global sparsity?
>
> We apologize, but we are not sure what you mean here. We apply the TopK sparsity activation function only to the active expert. Do you mean examining the number of features that would activate if we did not apply the TopK function, and comparing the chosen and non-chosen experts? We are happy to run this experiment if you clarify!
>
> If our comments resolve your concerns, we would appreciate it if you would consider raising your score!
>
> [1] Towards Monosemanticity: Decomposing Language Models With Dictionary Learning, Bricken et al.
>
> [2] Improving Dictionary Learning with Gated Sparse Autoencoders, Rajamanoharan et al.
>
> [3] Scaling and evaluating sparse autoencoders, Gao et al.

---

> > ### Comment · Reviewer_GyB2 · 2024-12-01
> >
> > Thanks the authors for the response.
> >
> > > Do you mean examining the number of features that would activate if we did not apply the TopK function, and comparing the chosen and non-chosen experts?
> >
> > I meant to ask about the overlap between supports of chosen and non-chosen experts after applying TopK function, just to get an idea. You do not need to run more experiments.
> >
> > Thanks for clarifying my concerns and questions. I changed my score.

---

> > > ### Author Response · Authors · 2024-12-02
> > > **Response**
> > >
> > > We are glad that our response helped to clarify your concerns. Thank you for your thoughtful feedback, which has significantly improved the quality of our paper.
> > >
> > > > I meant to ask about the overlap between supports of chosen and non-chosen experts after applying TopK function, just to get an idea. You do not need to run more experiments.
> > >
> > > Thank you for this excellent suggestion. We did not measure this, but will consider including it in a future revision.

---

### Author Response · Authors · 2024-11-28
**Overall Response**

Thank you to all the reviewers for their detailed comments! We have responded to all the reviews and have updated our submission with the suggested changes (highlighted in blue). Please let us know if any of your concerns have not been addressed.

A common concern amongst reviewers was the decreased performance of Switch SAEs in the fixed-width setting. In the right subplot of Fig. 1, we show that the gap between TopK and Switch SAE performance at a fixed width shrinks as we scale the number of parameters. Thus, for large-scale experiments, this gap may be imperceptible; since this is the regime in which efficient training is most useful, we believe that this is not a significant weakness of Switch SAEs.

In contrast, our Figure 3 fixed width results focus on the relationship between sparsity and performance at a single width of around 24K features, which is a regime where Switch SAEs are outperformed by TopK. We apologize for not making this more clear initially and have added additional text to our submission to help clarify this in Section 4.1 and in the caption of Figure 1.

---

### Meta-Review · Area_Chair_1gdE · 2024-12-21

**Metareview:**

The paper introduces an improvement to sparse autoencoders (SAEs) which are a (currently) popular set of tools used in LLM interpretability. SAEs suffer from the fact that the encoder is dense, and therefore compute-heavy; the contribution of this paper is to sparsify the encoder using a "switch" / routing layer that selects one of several smaller encoders/"experts". This enables improvements; for iso-FLOPs, the proposed switch SAEs are easier to train and obtain lower reconstruction cost. The features obtained by the resulting SAEs achieve comparable results on interpretability tasks as standard SAEs.

The paper is very well-written. Although the idea is a straightforward composition of existing tools (SAEs + switch layers), it is very well executed and may spur further work.

The drawback, as pointed out by a couple of reviewers, is that the number of features blows up, which may affect human-interpretability. To this point, the authors make the reasonable case that reconstruction error affects downstream tasks (including interpretability!), so on net this tradeoff might be worth it. I'd encourage the authors to probe this point a bit more in detail while preparing the final version.

**Additional Comments On Reviewer Discussion:**

All reviewers were generally positive about this paper. Some concerns were raised regarding potential applicability to interpretability research (as pointed above) but the authors responded satisfactorily.

---

### Decision · Program_Chairs · 2025-01-22

Accept (Poster)